# Sustainable and Elastic Carbon Aerogel by Polydimethylsiloxane Coating for Organic Solvent Absorption and Potential Application for Sensors (Infections, Environmental, Wearable Sensors, etc.)

**DOI:** 10.3390/ma16134560

**Published:** 2023-06-24

**Authors:** Youngsang Chun, Eui-Hwa Kim, Chae-Seok Lee, Hojong Chang, Chan-Sol Kang

**Affiliations:** 1Department of Advanced Materials Engineering, Shinhan University, 95, Hoam-ro, Uijeongbu-si 11644, Republic of Korea; chunys@shinhan.ac.kr (Y.C.); phdhippo@shinhan.ac.kr (E.-H.K.); 2KAIST Institute for Information Technology Convergence Integrated Sensor Team, KAIST, Daejeon 34141, Republic of Korea; quarry@kaist.ac.kr

**Keywords:** carbon aerogel, PDMS, waste paper, coating, microwave heating

## Abstract

Carbon aerogel is a promising material in various applications, such as water treatment, insulators, catalysts, and sensors, due to its porosity, low density, conductivity, and good chemical stability. In this study, an inexpensive carbon aerogel was prepared through lyophilization and post-pyrolysis using waste paper. However, carbon aerogel, in the form of short belts, is randomly entangled without a crosslinking agent and has weak mechanical properties, thus limiting its applications, which would otherwise be various. In this paper, a novel strategy is proposed to fabricate a PDMS-coated carbon aerogel (Aerogel@PDMS). Benefiting from microwave heating, precise PDMS coating onto the carbon frame was able to be carried out in a short amount of time. PDMS coating firmly tied the carbon microstructure, maintaining a unique aerogel property without blocking its porous structure. FE-SEM, RAMAN, XPS, and FT-IR were all used to confirm the surface change in PDMS coating. Compressible stability and water contact angle measurement showed that Aerogel@PDMS is a perspective organic solvent absorbent due to its good resilience and its hydrophobicity, and, as a result, its organic solvent absorption capacity and repeated absorption were evaluated, ultimately suggesting a promising material in oil clean-up and pollution remediation in water. Based on our experimental results, we identified elastic carbon aerogels provided by a novel coating technology. In the future, then, the developed carbon/PDMS composite can be examined as a promising option for various applications, such as environmental sensors, virus sensors, and wearable sensors.

## 1. Introduction

The development of petroleum processing technology has led to remarkable economic growth, but it has caused problems with the vast amounts of wastewater containing oil or organic solvents, whether in soil, rivers, or oceans. Such wastewater causes serious ecological problems, and this obviously affects mankind. There are several methods available for wastewater treatment, including biological remediation, physical treatment, and chemical treatment. Among them, physical adsorption has received attention due to its easy operation, its low cost, and its high efficiency [1,2,3,4]. Several materials, such as metal-organic frame [5], zeolite [6], fiber [7], and sponge [8], have been studied for both their desirable performance and their practical application. The scientific community has also been attracted to carbon aerogel as a contaminant detector, as this can prevent social problems arising from forms of contamination [9]. However, absorbents for oil or organic solvents require the ability to store contaminants inside their structure. Research has focused on carbon aerogel as an oil/organic solvent-absorbing material (OAM) because of its chemical robustness, its porous structure, and its low density [10]. However, despite these advantages, improvements still need to be made to the manufacturing of carbon aerogels, including the long production time, the utilization of petroleum-based chemical monomers, and the harsh conditions under which they are made [11].

Abundant biomass residues after various human activities are produced worldwide. Recent studies have been interested in the potential of biomass due to its structure, which is composed of sugars. Heat treatment can convert these sugars in biomass into carbon aerogel, which is a porous structure, depending on its initial shape. Therefore, biomass is a sustainable, reliable, and eco-friendly source of carbon aerogel precursors [12]. Among the various biomass residues, paper is the major biomass residue in Korea. This is because Korea is the world’s ninth-largest paper producer, and the country generates 48,990 tons of waste paper per day [13]. The development of carbon aerogels derived from waste paper is required to manage municipal solid waste, then, and to supply advanced materials for water treatment. According to several studies, carbon aerogel derived from waste paper has been developed as an oil–water separator, but the carbon aerogel converted from waste paper lacks any crosslinking between the biomass precursors, resulting in both poor shape retention and low mechanical strength [14].

Cyclic utilization is a crucial consideration for developing competent OAMs [15]. In particular, when regenerating OAMs, physical compression is required for the removal of absorbed liquid materials. Excellent mechanical strength and elasticity thus become essential features of good OAMs. Dong et al. [16] reported the addition of natural additives inside the wavy carbon layer in order to provide adhesion between the backbones. The prepared structure contributed to supercompression (with a strain of 95%) and super-elasticity (with retention after 500 cycles at a strain of 50%). Outstanding compressibility and elasticity were measured under external force, but the performances of both continued to decrease compared to the initial state under cyclic stress. In addition, the main skeleton, composed of graphene oxide, causes problems of high material cost and low productivity. Yi et al. [17] demonstrated a novel design of a spring-like structure of chitosan aerogel by directional freezing technology, and the unique morphology contributed to fast elastic recovery and, also, the mechanical stress being maintained for 300 cycles at a repeated compressing strain of 60%. The prepared aerogel is regarded as a green material in terms of the utilization of chitosan, but chitosan is obtained by several processes of strong acid and base chemicals. It is, therefore, necessary to develop low-cost and renewable resources with adequate properties for OAM.

Polydimethylsiloxane (PDMS) is a silicone polymer widely used in the fields of coatings, adhesives, sensors, pharmaceuticals, cosmetics, and adsorbents, and this is due to it having many advantages, such as elasticity, low toxicity, chemical stability, hydrophobicity, and commercial availability [17,18]. Due to the thermosetting property of PDMS, several researchers have recently paid attention to a surface coating agent or hydrophobic substrate material for preparing OAM. However, porous PDMS requires both elaborate porogen control and heat control during polymerization [19,20]. 

In this study, we prepared a carbon aerogel converted from waste paper, which is one of the municipal solid wastes regarded as biomass [21]. However, carbon aerogel composed of short belt-type carbons is fragile against external force, and the limitations of the applications can thus be predicted. In this study, then, elastic PDMS coating onto a carbon aerogel frame (Aerogel@PDMS) was carried out to bind each carbon backbone. During PDMS coating, microwave-assisted heating, which created local heating, was used to uniformly and precisely coat the carbon frame without blocking pores. PDMS contributes to the construction of compressibility and resilience. The characteristics of PDMS are also advantageous for absorbing hydrophobic organic solvents. Under precise and thin PDMS coating, Aerogel@PDMS is successfully designed to repeatedly use OAM by the physical compression method. The development of Aerogel@PDMS provides a green, sustainable, and renewable route for making carbon aerogel, but crucially, it also exhibits a technology that compensates for the mechanical properties of the fragile waste paper-based carbon aerogel.

## 2. Materials and Methods

### 2.1. Materials

Waste paper, as an inexpensive precursor for carbon aerogel, was collected from a lab. Polydimethylsiloxane (PDMS, Sylgard 184 A, and curing agent) and all chemical reagents (acetone, chloroform, cyclohexane, dichloromethane, ethanol, hexane, and toluene) were purchased from Sigma-Aldrich (Seoul, Republic of Korea), and they were used without further purification.

### 2.2. Preparation of Carbon Aerogel from Waste Paper

A total of 1 g of waste paper was immersed in 1 L of distilled water, and it was then kept agitated at 80 °C until the tangled cellulose pulp disappeared. To obtain dispersed clean pulp, impurities (ink, dye, etc.) floating on the upper side of the mixture were filtered out during the agitation. A micro-belt type of pulp was collected in a desired shape container, and a lyophilization process was performed to obtain the precursor of carbon aerogel. The prepared precursor was transferred into the furnace (PTF-12303, U1-Tech). It was heated up to 900 °C at a heating rate of 5 °C min^−1^ and then kept at 900 °C for 2 h in a nitrogen atmosphere.

### 2.3. Preparation of PDMS Coated Carbon Aerogel (Aerogel@PDMS)

The prepared carbon aerogel was immersed in hexane/PDMS (a mixture of backbone pre-polymer and curing agent (10:1 ratio)) solution for 1 h. The concentration of PDMS (10 wt%) was prepared based on the weight of hexane. The microwave heating process was performed with a 50% input power of 700 W for 10 s by microwave oven (KR-L202BGP, Daewoo, Seoul, Republic of Korea). It was repeated 10 times in order to carry out uniform polymer coating on the surface of the aerogel. After the microwave process, Aerogel@PDMS was immersed in hexane in order to remove uncured PDMS using sonication, and it was then dried at 80 °C (Figure 1).

### 2.4. Characterization of Aerogel@PDMS

A field emission scanning electron microscope (FE-SEM) instrument (S-4800, Hitachi, Tokyo, Japan) equipped with an energy-dispersive X-ray spectroscope (EDAX, Pleasanton, CA, USA) was used to obtain the surface images and atomic distribution of the prepared materials. The samples for FE-SEM analysis were sputter-coated with Pt/Pd (E-1010, Hitachi, Tokyo, Japan). The Fourier transform infrared (FT-IR) spectroscopy (Frontier, PerkinElmer, Waltham, MA, USA) was used to obtain the surface property as PDMS coating on the carbon aerogel. X-ray photoelectron spectroscope (XPS, X-tool, ULVAC-PHI, Hagisono, Japan) analysis was carried out, and CasaXPS (version 4.1) software was employed to process the peak analysis. Spectra were obtained with a monochromatic Al Kα X-ray source (1486.6 eV of photons). The analysis chamber was maintained at a pressure of less than 7.5 × 10^−7^ Pa during the measurement. All binding energies were referenced to the neutral C1s peak at 285.0 eV in order to compensate for the surface-charging effects. Intensity ratios were converted into atomic compositions using the sensitivity factors provided by the manufacturer. Raman spectra of the prepared materials at 100–4000 cm^−1^ were recorded by Raman spectrometer (Lab-RAM HR Evolution, Horriba, Tokyo, Japan). Visual images of thermal conditions of a microwave reaction vessel were obtained by the thermal camera (GTC 400 C Professional, BOSCH, Renningen, Germany). The water contact angle (CA) of the material was measured using a contact angle goniometer (phx300, SEO, Busan, Republic of Korea). Lastly, compressible stability was obtained by a universal testing machine (UTM; model 4467, Instron, Norwood, MA, USA).

### 2.5. Absorption Capacity of Aerogel@PDMS

The absorption capacity (g/g) was calculated by Equation (1):k = ((m_f_ − m_i_))/m_i_(1)
where m_i_ is the weight of the initial state of the absorbent and mf is the weight of the saturated state of the absorbent. When measuring the weight of liquid contaminants, the prepared absorbent was immersed in organic solvent until it was saturated, and it was measured immediately after removing the absorbent from the beaker. To evaluate the reusability of the absorbent, Aerogel@PDMS was squeezed powerfully by hand, washed with ethanol using an ultrasonic cleaner, and then dried in a convection oven at 120 °C. The reusability test of the prepared adsorbent was performed 30 times.

## 3. Results and Discussions

### 3.1. Fabrication of Aerogel@PDMS via Microwave Heating

Based on the heating characteristics of microwave irradiation, hexane is used as a reaction medium in the process of PDMS coating at the surface of the carbon aerogel. It was tan δ that was the key factor of solvent selection. The loss factor is expressed as tan δ = ε″/ε′, where ε″ is the dielectric loss indicating induced heat converted from electromagnetic radiation, and ε′ is the dielectric constant that shows the capability of the polarizability of the molecule by an electric field. The values of the loss factor of solvents that had permanent dipoles were classified as high (tan δ > 0.5), medium (tan δ 0.1–0.5), and low (tan δ < 0.1), with a value over 0.1 effectively responding to microwave, thereby releasing rapid heat [22]. Hexane (tan δ ≈ 0.02) [23] was adequate to use as a medium under microwave irradiation, and it was homogeneously accessible to PDMS pre-polymer due to having similar solubility parameters [24]. In our experiment, the prepared reactor containing the carbon aerogel soaked in 10 wt% PDMS with hexane (B.P. 68 °C) was immediately measured after microwave irradiation (Figure 2).

The initial temperature of the reaction medium was about 25 °C after it was exposed to microwave irradiation, as mentioned in the experimental section above, and the hexane reached about 65 °C with the gas-releasing phase of hexane. Choi et al. experimentally confirmed PDMS layers of 10 nm under a curing temperature of 67 °C [25]. It was estimated that microwave heating sufficiently performed an ultrafast and localized PDMS curing. Carbon is a unique structure that conducts electricity and has a low resistance to the flow of electricity, and these properties make carbon a suitable material to be heated by microwave irradiation. The microwave energy causes ions or electrons inside the carbon molecule to vibrate, and this increases the heat inside the molecule and also heats the carbon [26,27]. This phenomenon causes instantaneous heat at the surface of the carbon. The carbon structure was confirmed by Raman spectroscopy in Figure 3.

The temperature (Figure 2b) of microwave-less responsive hexane was increased by short irradiation. The condition is considered sufficient to produce selectively polymeric coatings.

### 3.2. Characterization of Aerogel@PDMS

FE-SEM images showed the surface changes in the carbon aerogel by PDMS coating (Figure 3). The carbon aerogel was a porous and well-interconnected 3D network structure consisting of arbitrarily entangled micro belt-type with a rough surface, obtained by the pyrolysis of waste paper (Figure 3a). After fast microwave irradiation, a smooth surface was observed with the PDMS layer at a narrow responsive surface region (Figure 3b). The PDMS layer onto the carbon aerogel could be cured by quick localized heating. Furthermore, randomly entangled carbon belts were strongly tied by PDMS coating, which was caused by releasing the highly reactive heat that was induced by microwave irradiation (Figure 3c,d). It is expected that the PDMS coating on the carbon aerogel will enhance both the compressible property and the reusability of the carbon aerogel. Elementary mapping (C, Si, and O) images of aerogel and Aerogel@PDMS were clearly shown in Figure 3e,f. C atom (red) was well illustrated as being the same as the 3D micro-belt type networks of SEM images (Figure 2e, first image). After PDMS curing, as shown in Figure 3f, the distribution of the Si atom (cyan) and O atom (yellow) followed that of the carbon backbone attributing to the PDMS layer at the surface of the carbon aerogel. Compared to Figure 3e, before PDMS coating, the red image of the carbon atom only appears without the cyan and yellow images.

Raman spectra of carbonaceous materials typically showed the two major peaks of the D-band (~1330 cm^−1^) and G-band (~1580 cm^−1^), where the presence of the D-band and G-band indicated that polyaromatic hydrocarbons and graphitic carbons existed, respectively (Figure 4a) [28]. A typical D peak, G peak, and 2D peak (2690 cm^−1^) of the carbon aerogel were observed, which corresponded to the pyrolysis of cellulose from waste paper. Characteristic peaks of Aerogel@PDMS were observed at 490, 707, 2905, and 2965 cm^−1^, associated with the symmetric stretching of Si–O–Si, the symmetric of Si-C, and symmetric/asymmetric vibration of CH_3_ [29,30]. The elemental analysis assessed by XPS confirmed that an atomic ratio of C, O, and Si changed the PDMS coating, as shown in Figure 4b.

An increase in Si and O atoms was observed, which indicated the PDMS coating on the carbon aerogel. Pristine carbon aerogel and Aerogel@PDMS were investigated by XPS analysis (Figure 4c). The XPS survey spectrum of carbon aerogels showed the main chemical components of carbon and oxygen as C1s and O1s peaks. Thin PDMS curing onto the carbon aerogel was confirmed by the appearance of sharp Si2s and Si2p peaks at 101.0 eV and 152.0 eV in the XPS spectrum of Aerogel@PDMS [31]. In addition, the increasing intensity of the O1s peak was caused by the chemical structure of the siloxane group. Figure 4d shows the FT-IR spectrum of waste paper (gray), aerogel (black), pristine PDMS (blue), and Aerogel@PDMS (cyan). A spectrum of the waste paper appeared at a broad peak at 3000–3400 cm^−1^, representing the hydroxyl group. Compared with a spectrum of carbon aerogel, the peak indicated that the waste paper composed of cellulose had been converted to carbon aerogels. A spectrum of pristine PDMS showed prominent peaks of CH_3_ stretching and CH_3_ deformation vibration at 2963 cm^−1^ and 1280 cm^−1^, respectively, the peaks of which were also observed in the Aerogel@PDMS [32]. The Si–O–Si multicomponent peak and other Si(CH_3_)_2_ rocking representing the PDMS backbone were investigated at 930–1200 cm^−1^ and 785–815 cm^−1^, respectively [33]. All analyses confirmed that PDMS coating onto a carbon frame is carried out well by microwave-assisted heating.

Figure 5 shows the surface chemical property of the carbon aerogel and Aerogel@PDMS.

The measurement of water contact angles indicated that each surface of the prepared materials was hydrophobic, showing water drops in various pH maintained on the surface. However, the water contact angle (Figure 5a inset and Figure 5b inset) of PDMS aerogel was higher than that of the carbon aerogel, which showed that PDMS coating became a more hydrophobic surface. Different phenomena, as shown in Figure 5b,d, confirmed PDMS coating inside the carbon aerogel.

The compressible stability of oil/water absorbents is an important property for practical operation and reuse. Figure 6 shows the cyclic stress–strain curves of Aerogel@PDMS at 60% of maximum stress.

Bare carbon aerogel is composed of randomly entangled short carbon fibers, which are completely broken down by external force. However, the curves of Aerogel@PDMS during 30 cycles are observed to be similar to the initial curve due to PDMS coating onto the carbon structure. Elastic PDMS coating contributed to the resilience, providing good stability for repeat use and long-term usage. It is expected that PDMS coating onto a carbon frame can be developed for various applications in the field of porous carbon-based novel material.

### 3.3. Aerogel@PDMS for Organic Solvents Absorbent and Regeneration Test

The absorption capacity of Aerogel@PDMS for organic solvents is evaluated for the potential possibility of using it as OAM. The mass absorption capacity was calculated as the mass ratio between the absorbed liquid and the dried aerogel. Figure 7a shows the absorption capacities for various organic solvents in the range of 4.2 to 8.7 g/g.

The sorption capacities for chloroform, dichloromethane, toluene, acetone, and cyclohexane were 8.7, 7.2, 5.8, 4.6, and 4.2.g/g, respectively. Various sorption capacities depend on the density of the tested liquids [34]. Such an absorption capacity demonstrated that PDMS outside the carbon frame performs hydrophobic solvent absorption. Moreover, the recycling performance of Aerogel@PDMS was examined for practical applications, and there was no significant change in absorption capacities over the 30 cycles for all of the solvents. It is thought that elastic PDMS wrapping the carbon aerogel enhanced resilience by binding all of the carbon frames. Constant absorption capacities also indicated that a novel coating is completely carried out inside the carbon aerogel. Compared to the absorption capacities of untreated carbon aerogel, about 45% of capacity decreases were measured in all of the tested solvents (Figure 7b). This tendency is due to the increase in the weight of the absorbent by the PDMS coating on the carbon aerogel surface. Further work is still needed to investigate the optimal ratio of PDMS coating that will maintain the resilience of carbon aerogels and also exhibit competent absorbency, and this can be expected with expanded research into Aerogel@PDMS-based sensors, insulators, and catalysts [35,36,37].

### 3.4. Comparison of Different Aerogels 

In recent decades, several studies have been conducted to develop aerogels for organic solvent removal. Aerogel, which has the advantages of porosity and is lightweight, is suitable for absorbing hydrophobic organic solvents due to its surface characteristics. Table 1 summarizes the characteristics of biopolymer, waste plastic, and fiber-based aerogel for density, contact angle, absorption, and reusability. Pawar et al. developed waste plastic-based aerogel using a crosslinking agent [38]. Through the synergy of the additives, this material had excellent mechanical properties. The contact angle of 145.9 degrees exhibited its absorption capacity for hydrophobic materials. However, a 70% reusability rate was shown in 10 evaluations. Cellulose-based aerogels, widely known as biopolymers, were developed by Sai et al. and Chen et al. [39,40]. Both materials showed contact angles of 146.5 degrees and 148.7 degrees, respectively, and they were suitable for the removal of hydrophobic solvents. In addition, both materials showed similar performance to the initial state after the 10-time reuse evaluation. Li et al. and Loh et al. developed aerogels using biomass and waste fibers as a green chemistry strategy [41,42]. Contact angles of 148 degrees and 138 degrees were confirmed for the two materials, respectively, and were suitable for re-moving hydrophobic solvents. However, it shows limitations in reusability evaluation and requires improvement, such as crosslinking or coating for good resilience. Aerogel@PDMS is an aerogel developed using waste paper, and it is a material with excellent resilience because it is coated with PDMS. The contact angle was shown at 125 degrees, which was suitable for hydrophobic solvent absorption, and results similar to the initial ones were confirmed in the absorption evaluation repeated 10 times. Based on the comparison, it was confirmed that aerogel with a hydrophobic surface could absorb organic solvents, and that aerogel-type absorbents are required to undergo post-treatment for reusability.

## 4. Conclusions

In this work, Aerogel@PDMS was obtained by a thin PDMS coating on carbon aerogel via microwave heating. First, the prepared carbon aerogel was converted from waste paper, suggesting a sustainable biomass utilization route. The PDMS coating was then carried out by local heating that occurred from microwave irradiation around the carbon frame. The coating technique is a novel method to compensate for the weak mechanical properties of biomass residue-based carbon aerogel without additional crosslinkers and blocking pores. According to organic solvent absorption capacity and repeated compressible stress–strain curves of Aerogel@PDMS, PDMS corresponded to an adequate thermosetting polymer due to its hydrophobicity and its elasticity for fabricating OAM. Consequently, Aerogel@PDMS shows a water contact angle of 125 °C and an absorption capacity of 4.2–8.7 g/g for various organic solvents. The sorption rate during the repeated 30 cycles also stayed above 99% due to its stable structural resilience. Accordingly, microwave-assisted thin PDMS coating on the carbon frame suggests that elastic carbon aerogel is a promising material that can be utilized in the field of wastewater treatment. 

As we have described and shown in this paper, Aerogel@PDMS has been well developed for absorbent purposes. Now, we anticipate the development of an advanced carbon/PDMS composite for various applications, such as environmental sensors, virus sensors, and wearable sensors, since carbon material has received attention due to its outstanding electrical properties, and PDMS coating has shown both elasticity and resilience. Carbon materials, such as carbon nanotubes and graphene, have received a large amount of attention due to their active sensing layer in terms of reactivity and sensitivity. However, the limitations in large-scale production using these carbons should be improved. This is especially important in view of the fact that this study presented carbon aerogel from waste paper as a green strategy, and it is thus important to find ways of obtaining an easy and inexpensive precursor of carbon material. Through upcycling, indeed, a cost reduction effect from an environmental point of view should be expected, as should a reduction in manufacturing cost compared to other materials. When commercialized, indeed, great economic effects can be anticipated.

## Figures and Tables

**Figure 1 materials-16-04560-f001:**
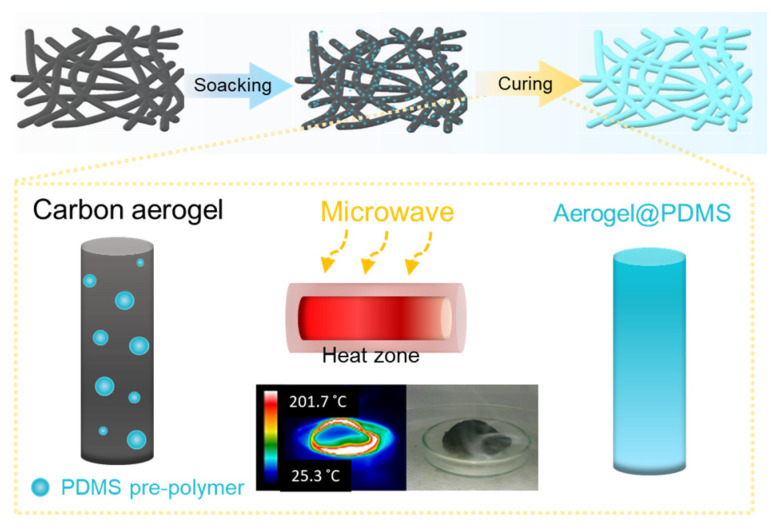
Schematic illustration of the preparation of Aerogel@PDMS.

**Figure 2 materials-16-04560-f002:**
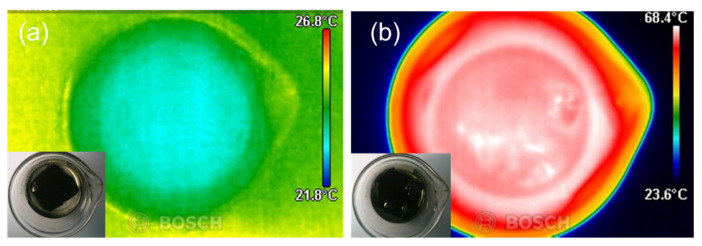
Visualization of the thermal condition of carbon aerogel soaked in hexane: (**a**) initial state, (**b**) after microwave irradiation.

**Figure 3 materials-16-04560-f003:**
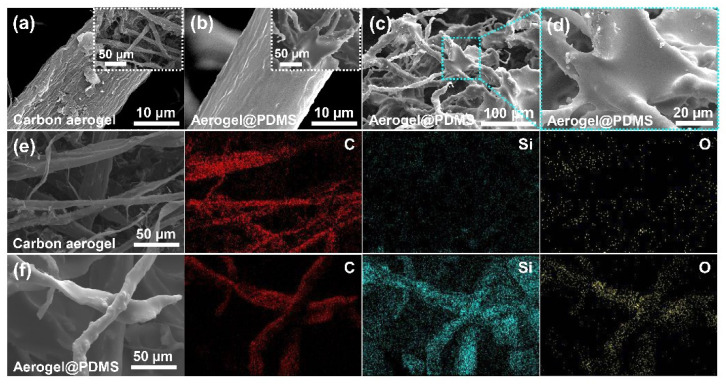
Fe-SEM images of carbon aerogel and Aerogel@PDMS: (**a**) the surface of carbon aerogel; (**b**) after thin PDMS coating on carbon aerogel; (**c**) PDMS-tied micro-belt type carbon fibers in Aerogel@PDMS; (**d**) magnification of the PDMS-tied area in Aerogel@PDMS; (**e**) the overall appearance of carbon aerogel, and elementary mapping of C (red), Si (cyan), and O (yellow), respectively; and (**f**) overall appearance of Aerogel@PDMS, and elementary mapping of C (red), Si (cyan), and O (yellow), respectively.

**Figure 4 materials-16-04560-f004:**
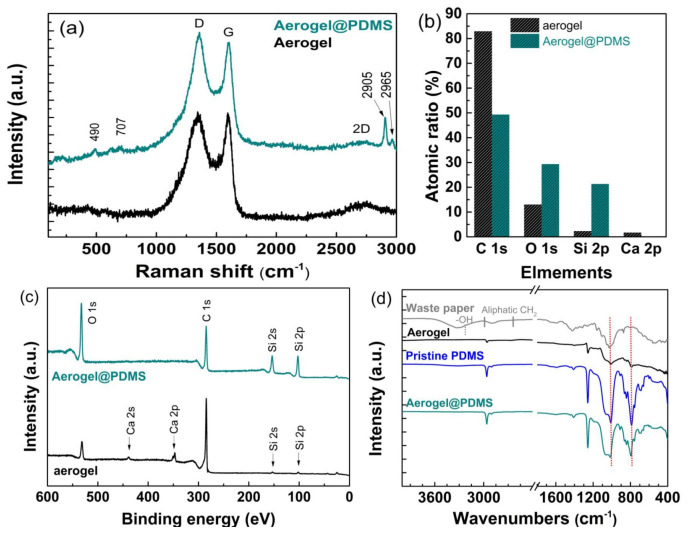
(**a**) Raman spectra of carbon aerogel Aerogel@PDMS. (**b**) Atomic ratio of carbon aerogel and Aerogel@PDMS. (**c**) XPS spectra of carbon aerogel and Aerogel@PDMS. (**d**) FT-IR spectra of waste paper, carbon aerogel, pristine PDMS, and Aerogel@PDMS.

**Figure 5 materials-16-04560-f005:**
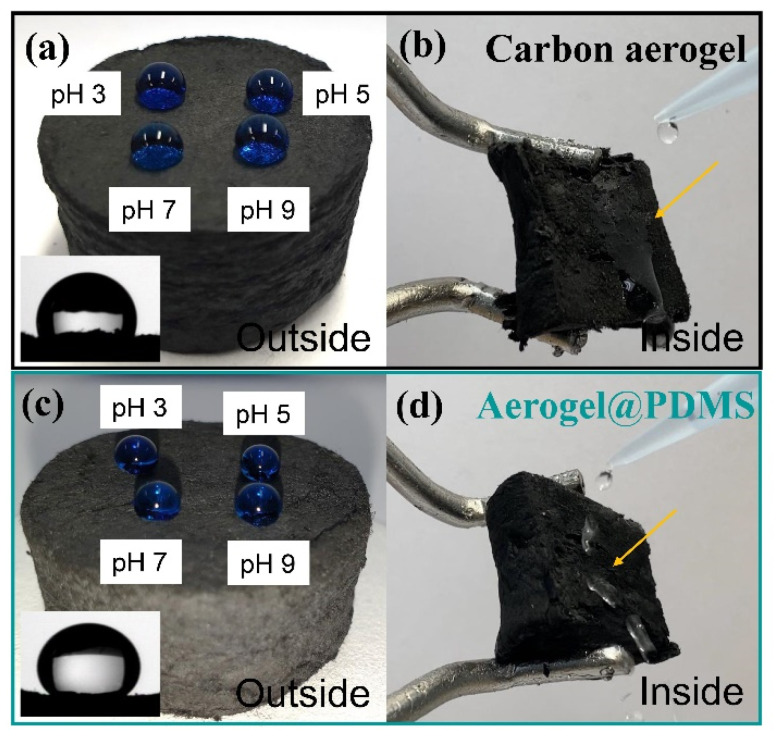
The image of water droplets in various pHs onto the surface of carbon aerogel (**a**) and the water contact angle (inset). The dripping of water inside of carbon aerogel (**b**). The image of water droplets in various pHs onto the surface of Aerogel@PDMS (**c**) and water contact angle (inset). The dripping of water inside of carbon aerogel (**d**).

**Figure 6 materials-16-04560-f006:**
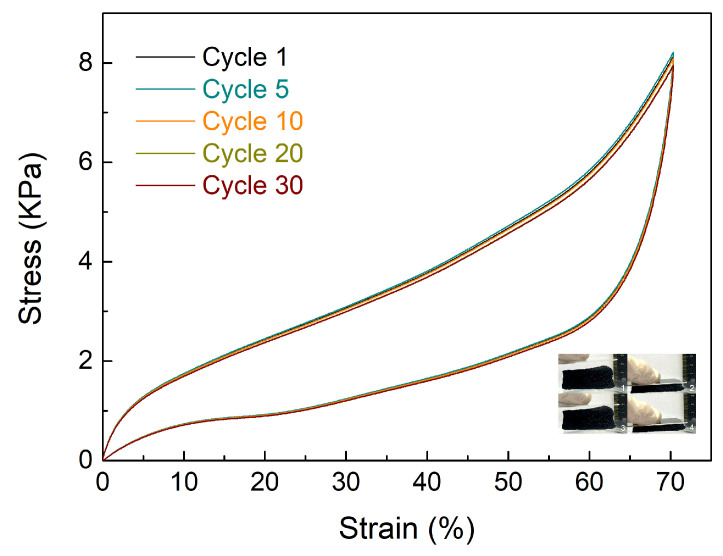
Compressive stress–strain curves of Aerogel@PDMS. Inset: visual representation depicting the prepared sample.

**Figure 7 materials-16-04560-f007:**
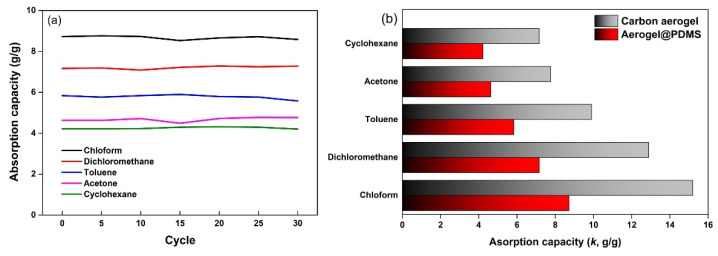
(**a**) Absorption capacity during 30 cycles and (**b**) comparison of absorption capacity between carbon aerogel and Aerogel@PDMS.

**Table 1 materials-16-04560-t001:** Various aerogels in organic solvent absorption applications.

Aerogel	Density (mg/cm^3^)	Water Contact Angle (Degrees)	Absorption Capacity (g/g)	Reusability (during 10 Times)	Ref.
Plastic waste-based aerogel	311	145.9	27.43	70%	[38]
Cellulose aerogel	6.77	146.5	185	99%	[39]
Alginate/cellulose aerogel	-	148.7	34 times	99%	[40]
Bagasse-based aerogel	47.3	148	38.76	50%	[41]
Wool waste fiber-based aerogel	4	138	36.2	-	[42]
Aerogel@PDMS	59	125	8.7	100%	This work

## Data Availability

MDPI Research Data Policies.

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
