# Peer review of "Sustainable and Elastic Carbon Aerogel by Polydimethylsiloxane Coating for Organic Solvent Absorption and Potential Application for Sensors (Infections, Environmental, Wearable Sensors, etc.)"

_materials, 2023, doi:10.3390/ma16134560_

Round 1

Reviewer 1 Report

The papers describes fabrication of carbon gel using waste paper and improvement of its mechanical properties by coating PDMS. Overall, this paper is well written and organized.  It is found that this carbon gel has a potential in wastewater treatment as a absorbent material. Overall, this paper is well written and organized. In my opinion, the manuscript is suitable for publication in Materials, after the authors have addressed the following comments and questions:

1. Last sententence in abstract should be modified (In our point of view?)

2. Rephrase the sentence, line 84 Also, intrinsic hydrophobic property of PDMS is expected to absorbing organic solvent

3. There is missing a water contact angle values 3. I suggest to authors to give a litertaure review about different aerogels (Table)

4. In Fig 7a there is missing a label for x and y axis.

5. In order to clarify the results, the authors need to demonstrate the oil absorption kinetics.

6. In conclusion, line 287 gel@PDMS shows a water contact angle of 000 degrees. Is this a mistake or? Also, the authors hadn't give any experimental result regarding to sensor application, I suggest to rephrase the the title of manuscript and erase potential application for sensor.

Reviewer 2 Report

The study entitled Sustainable and Elasitc Carbon Airgel by PDMS Coating for Organic Solvent Absorption and Potential Application for

Sensor, is interesting and fits within the scope of the journal, however some changes can be made aiming at a better quality of it.

Please in the introduction describe the importance of detection and sensor development beyond detection. In this sense, it is important to apply technologies aimed at removing these compounds from the environment, such as adsorption technology. Below are some articles that may help in this regard.

https://www.sciencedirect.com/science/article/pii/S221471442300140X

https://www.sciencedirect.com/science/article/pii/S0167732223004804

In the introduction, the authors should make it clear what is new about the study.

In conclusion, the limitations of the study and future perspectives should be described.

English must be revised.

Sensor cost analyzes can be included in the study.

Minor editing of English language required

Reviewer 3 Report

Dear authors, thank you for your submission and congratulations on the high importance of your article.

The manuscript is well-structure and provides a clear methodological and results presentation.

Consider including some limitations of your study in the discussion topic.

Abstract: “In our point of view, …” I do not agree with this mention. Consider objectively and accurately rewriting the results observed in your study.

Conclusion: “… a water contact angle of 000 degrees”. Is this a correct information?

Reference 3, 5, 20 need to be corrected.

Discussion: Attribute real impacts to each result found, favoring the reader’s clarity to justify being a promising product, as stated in the conclusion.

Reviewer 4 Report

In this work, carbon aerogel was prepared from waste paper and then coated with a thin layer of PDMS in order to enhance mechanical properties. Then the material was tested for absorption capacity of different organic solvents. The composite aerogel/PDMS was characterized by different methods, which are adequately described. Presentation is clear and illustrations are adequate. Minor spelling mistakes should be checked.

Comments:

1. What is the estimated density of pristine aerogel and PDMS-coated aerogel?

2. The paper presents absorption data for clean small-molecule solvents and claims the possibility of oil cleanup. Did you test absorption capacity and cycling for oil and similar materials?

Minor spelling mistakes should be checked. For example, line 168 "Because Carbon is a unique structure".

Round 2

Reviewer 1 Report

I am not completely satisfied with the author’s responses to my questions/issues raised in my initial review.  I recommend that the revised paper can be accepted with very minor revisions.

Applications in sensors shoud be mentioned only in conclusion. Since there is no any experimental study regardiing to sensors applications, the part of Title of manusccript: and Potential Application for Sensors (infections, environmental, wearable sensors, etc.) need to be erased. 

Revise conslucion-some sentences are mentioned twice and it is confused to follow what is described.

 I recommend a professional round of language editing before the paper is published. 

Reviewer 2 Report

The authors provided a clear and complete review accepting all suggested changes. Therefore, the manuscript can be accepted for publication.

Author Response

I am particularly grateful for the clear and complete review you provided for our manuscript. Your insightful suggestions and constructive feedback have greatly improved the quality of our work. It is evident that you invested considerable time and effort in thoroughly reviewing our manuscript, and we are truly grateful for your meticulous attention to detail.